# Discussion of the Segregation and Low Hardness of Large-Diameter M3 High-Speed Steel Produced by Spray Forming

**DOI:** 10.3390/ma16020482

**Published:** 2023-01-04

**Authors:** Jihao Liu, Hongxiao Chi, Huibin Wu, Dangshen Ma, Jian Zhou

**Affiliations:** 1Special Steel Department of Central Iron and Steel Research Institute (CISRI), Beijing 100081, China; 2Collaborative Innovation Center of Steel Technology, University of Science and Technology Beijing, Beijing 100083, China

**Keywords:** spray-formed large-diameter high-speed steel, segregation, low hardness, solidification model, transformation of microstructure

## Abstract

As an advanced near-net-shape processing method in which directly preformed, semi-finished products are created from liquid metals, spray forming has become popular in the development and application of new materials and is supporting industrialization. However, as investigated in this work, the problems of segregation and low hardness exist in the actual industrialization process, particularly for large-diameter M3 high-speed steel. It was here found that the annual ring segregation morphologies were mostly distributed from the edge to 1/2*R*, with a large number of stripes primarily enriched in C, Mo, and Cr elements, and the degree of segregation was mild. The ring segregation was located at the 1/2*R* position, where the main elemental enrichments were C, W, Mo, Cr, and V, and the segregation degree was severe. The formation of segregation during deposition is described based on an equilibrium solidification model. A slow cooling rate and heat dissipation from the surface to the inside were judged to be the main factors causing segregation and changes in the carbide morphology. In terms of hardness, with the increase in the quenching temperature to 1230 °C, the tempering hardness increased significantly. The analysis shows that a faster cooling rate in the atomization stage caused the solidified droplets to exhibit rapid solidification characteristics, and there was a higher proportion of MC carbide in the deposited billet. MC carbides cannot be fully dissolved using the conventional heat treatment process, which decreases the C, Cr, Mo, and V contents in the solution and, thus, reduces the secondary hardening capability. The findings show that, when the spray forming process is used to prepare large-diameter materials, it should not be considered a rapid solidification technology simply because of its atomization stage. Moreover, more attention should be paid to the influence of microstructure transformation during atomization and deposition.

## 1. Introduction

Due to its excellent wear resistance and red hardness, high-speed steel (HSS) is widely used in large machining tools, machinery manufacturing, and other fields [1,2]. To meet the developing needs of modern processing technology and market demand, more alloying elements, such as C, W, Mo, Cr, V, and Co, are added to HSS to further improve its strength and wear resistance and to achieve high performance [3,4]. However, problems such as the presence of coarse carbides caused by a high alloying element content have a negative effect on the service and process performance of HSS. These problems are difficult for the traditional casting and forging (CF) method to overcome [5,6,7]. Improvements in the cooling rate can effectively relieve the segregation and refine the microstructure of HSS, and uniform and dispersed carbides can greatly improve its performance. Therefore, powder metallurgy (PM) has become an important way to produce high-quality HSS [8,9].

With current societal and technological progress, problems such as pollution and energy consumption have attracted increasing attention, and greenness and sustainability are the pursuits of the entirety of human society [10]. The production and processing processes for iron and steel materials are capital-intensive [11]. Thus, reducing energy waste and equipment costs will become mainstream ways to develop the steel industry. Figure 1 presents three conventional smelting processes for tool steel production. In contrast to the electroslag remelting (ESR) and PM processes, the spray forming (SF) process eliminates the remelting and conversion steps, which are characterized by serious pollution, and reduces the cost of expensive equipment. SF has therefore become a key technology in the adaptation to current developments. SF mainly involves atomization and deposition, during which the metal liquid flow is broken into droplets by high-speed atomization gas and then deposited onto the corresponding shape of the deposition plate to complete the forming. The properties of materials produced with SF are usually situated between those of materials produced by ESR and PM [12,13].

By using numerical simulation to adjust the process parameters and complete the production of materials [15,16], the problems of holes [17] and of the recycling of overspray powders have been solved [18], and SF is becoming increasingly more mature and widely used in the production of high-alloy steel [19,20]. However, there remain several problems with SF that are worthy of consideration, the first of which is the debate about whether it is a rapid solidification technique. Due to its uniform equiaxed microstructure, dispersed distribution of the second phase, and low degree of macrosegregation, and because its combination with atomization processes is similar to that of PM, it seems natural to consider SF a rapid solidification technology. However, Grant [21] and Zepon [22] have proposed that the solidification process in SF takes place under near-equilibrium conditions in an approximate equilibrium state, and the equiaxed microstructure is the result of the dendrite fragmentation mechanism and the homogenization of the temperature and composition caused by liquid flow. Second, under the same heat treatment, HSS prepared with SF usually exhibits lower hardness compared to when the same composition of HSS is prepared with CF and PM [23]. Low hardness inevitably leads to HSS being unable to satisfy service requirements in harsh environments; however, few relevant studies and discussions of this problem have been published. The third problem is segregation; e.g., centrifugal segregation [24] in the production of aluminum alloys and concentric circle segregation [25] in the production of X110CrMoVAl8 cold work die steel. In particular, concentric circle segregation often occurs during the production of large-diameter M3 HSS. However, there is not yet a relevant theoretical basis for the formation of segregation.

In this study, considering the segregation and low hardness mentioned previously, an M3 HSS bar with a diameter of 250 mm was prepared with SF, and relevant research was carried out. First, samples were selected for composition testing and microstructure analysis according to the various segregation morphologies in the macrostructure, and the formation of segregation was described based on an equilibrium solidification model. Then, to investigate the problem of low hardness, different heat treatments were designed to study the dissolution and precipitation behaviors of carbides during quenching and tempering, as well as the changes in hardness. Finally, based on the experimental results, the characteristics and advantages of SF are discussed from the perspective of the atomization and deposition of SF.

## 2. Materials and Methods

### 2.1. Material Production Process

The test material was prepared using twin-nozzle SF equipment and with a chemical composition balance of 1.11C-6.12W-5.29MO-4.06CR-2.82V-Fe, which is a traditional M3 HSS. To reduce harmful impurity elements, electroslag heating refinement (ESH) was introduced into the production stage of the SF. The process flow, as shown in Figure 2, was as follows: induction furnace smelting → ESH refining → SF. In the SF stage, N_2_ was used as the atomized gas and deposited on a substrate with a diameter of 500 mm. After spraying and deposition, the billet was annealed and finally left the factory in the form of rolled and softened annealed bars. The products were 250 mm in diameter and 2 m in length. Experimental material was taken from the 3/4 position in the bar height direction, and the thickness was 10 mm.

### 2.2. Chemical Composition

A nitric acid solution with a volume ratio of 1:10:10 was used for the etching of the macrostructure, and a camera was used to record the morphology. Then, considering the special segregation morphology of the macrostructure, samples #1, #2, and #3 were selected for composition testing. The size of sample #1 was 120 mm × 25 mm × 10 mm, and the composition changes from the edge to the center were measured with a SPARK-800 arc-spark direct reading spectrometer (NCS Testing Technology Co., Ltd., Beijing, China). The sample was tested at a total of 11 points every 10 mm from the edge to the center. The size of samples #2 and #3 was 25 mm × 25 mm × 10 mm, and these samples were selected according to the morphological characteristics of segregation. An ELIBSOPA-200 device (NCS Testing Technology Co., Ltd., Beijing, China) was used to characterize the element distribution, and the device parameters were as follows: a Continuum Surelite^TM^ III-10YAG laser(Continuum, Boston, MA, USA) was used with a wavelength of 1064 nm, a beam divergence angle of 0.6 mrad, an energy of 700 MJ, an excited spot diameter of 300 μm, a line spacing of 300 μm, and a scanning area of 231 mm^2^. The main elements considered in the composition test were C, W, Mo, Cr, V, and Fe.

### 2.3. Heat Treatment

Heat treatment samples were taken from the edge of the experimental material. The sample size was 15 × 15 × 10 mm. The selected quenching temperatures were 1130, 1150, 1180, 1200, 1230, and 1250 °C. Each quenching temperature was held for 20 min, and the samples were then cooled to room temperature with oil. After quenching, the samples were immediately tempered at 560 °C for 1 h and air-cooled to room temperature, and this process was repeated three times. The Rockwell hardness method was used at room temperature to measure the hardness resulting from different heat treatments. The hardness of each sample was measured six times, and the average hardness was calculated as the final result.

### 2.4. Characterization

The phase identification of the samples was performed using X-ray diffraction (XRD, D8 ADVANCE X, Bruker AXS GmbH, Karlsruhe, Germany) with Co Kα radiation. To examine the annealed microstructure, an optical microscope (OM, MEF-4M, Leica, Wetzlar, Germany) was used for microstructure observation. The morphology of carbides was observed with a scanning electron microscope (SEM, FEI Quanta650FEG, Hillsboro, OR, USA) equipped with an energy-dispersive spectroscopy (EDS) device. To investigate the quenched microstructure, quantitative carbide statistics were calculated for carbides with sizes greater than 0.4 μm to eliminate the interference of objective factors, such as pixels. Fifteen photos were taken of the samples in each quenching temperature group for carbide statistical analysis in Image-Proplus 6.0 software under the BSE mode (SEM). A field emission electron probe micro-analyzer (EPMA, JEOL JXA-8530F, Tokyo, Japan) was applied for elemental analysis. To examine the tempered microstructure, analyses of nano-precipitates and selected area electron diffraction (SAED) were conducted using transmission electron microscopy (TEM, JEM-2100, HEOL, Tokyo, Japan) at an accelerating voltage of 200 kV. Thin film samples for TEM were prepared by ion milling.

## 3. Results

### 3.1. Alloying Element Distribution and Microstructure Characteristics

As can be seen from the macrostructure of the experimental material displayed in Figure 3, two kinds of segregation morphologies were observed, the first of which was light gray with a discontinuous annual ring corrosion pattern. The annual ring segregation was mainly distributed from the edge to the 1/2 radius (1/2*R*) region. The color of the other was dark gray and it had a ring shape positioned at about 1/2*R*. The results exhibited in Figure 4a shows that severe composition segregation occurred at the ring segregation location corresponding to 1/2*R*, while the composition was relatively uniform at the other positions. The XRD patterns of the edge (#6), ring segregation (#5), and center (#4) exhibited no significant differences, as shown in Figure 4b. The change in the peak strength was due only to the different phase proportions. The microstructure mainly consisted of the α-Fe matrix and primary carbides (M_6_C, MC, and M_23_C_6_).

The composition and microstructure analyses of the discontinuous annual ring segregation of sample #6 were further carried out, as shown in Figure 5. It can be seen from Figure 5a that there was a slight enrichment of the C, Mo, W, and Cr elements in the segregation. For the microstructure analysis of annealed samples, long-term corrosion (5 min) was selected to improve the contrast difference between the carbides (white) and matrix (black), and the morphology of the carbides was observed more clearly in the OM image. In combination with the microstructure observations presented in Figure 5b–e, it can be seen that, compared to the normal regional microstructure, not only did severe carbide bands exist (Figure 5c) but the network-like carbide was significantly coarser in the segregation area (Figure 5d,e). The SEM images and EDS analysis results in Figure 5f,g reveal that the morphologies of the carbides in both the segregation region and the normal region were mainly divided into granular-like M_6_C carbides and strip/chain-like M_6_C + MC complex carbides. For tungsten-molybdenum HSS, the type and morphology of carbide formation are mainly determined by the cooling rate [26]. Under normal conditions, M_2_C and MC carbides with a network morphology are usually precipitated by eutectic reflection due to the higher content of carbon and alloying elements at the grain boundaries during solidification. In the subsequent annealing and forging process, M_2_C carbides exist in a metastable phase, and they are dissolved into granular-like M_6_C and MC carbides due to the action of alloying element diffusion and the surface curvature. However, it is difficult for some M_2_C carbides to decompose completely due to size factors. Therefore, the morphologies of the complex MC and M_6_C carbides still exhibited a strip- or chain-like appearance, which corresponded to eutectic M_2_C carbides.

The composition of sample #5 and the microstructures of samples #5 and #4 are presented in Figure 6. It can be seen from Figure 6a that the segregation of the C, W, Mo, Cr, and V alloying elements was serious in the ring segregation. From Figure 6b–g, it can be seen that the morphologies of the carbides from the edge (sample #6) to the center (sample #4) were obviously changed. The morphologies of the carbides changed from strip/chain to massive, and serious stacking of carbides in the segregation area was observed. Some of the carbides were distributed in the shape of fish bones (Figure 6c). The massive carbides were significantly coarser than those in the normal regions of samples #5 and #4 (Figure 6d,e). Based on the SEM images and EDS results presented in Figure 6f,g, the types of carbides were mainly massive M6C and fish bone-like M_6_C. From the perspective of the solidification rate, there was another extreme case of carbide formation; namely, the precipitation of carbides during solidification changed from M_2_C to M_6_C at a very slow cooling rate [27].

The morphology of primary carbides depends on the crystal structure [28]. The crystal structure of M_2_C carbide is a close-packed hexagonal structure, and three directions with a large growth rate—namely, <1010>, <1100>, and <0110>—are all in the {0001} plane. The carbide grows more quickly in the {0001} crystal plane during the eutectic transformation. Thus, the morphology of an M_2_C carbide is in the shape of a stripe. Regarding the M_6_C carbide, as in a face-centered cubic (fcc) crystal, the nucleus consists of eight {111} crystal planes, and the point direction is <100>. When a nucleus crystallizes in a liquid, its morphology is massive. The atom internal to <100> is the largest in an fcc crystal, so the nucleus grows more quickly along the sharp point, which results in the M6C eutectic carbide attaining a skeletal morphology. The cooling rate determines the type of carbides, as shown in Figure 5 and Figure 6. The change in the carbide morphology from a strip/chain shape to a block shape from along the edge to the center of the sample implies that there was a difference in the cooling rate along the diameter direction during solidification. The description of the SF solidification process and the formation analysis of segregation are presented in Section 4.1.

### 3.2. Microstructure and Hardness Changes during Heat Treatment

#### 3.2.1. Microstructure Evolution during Quenching

Figure 7 shows the XRD patterns of the samples after quenching at different temperatures from 1130 to 1250 °C. It can be seen from Figure 7a that the microstructures after quenching at different temperatures were mainly composed of primary M_6_C and MC carbides, α-Fe (martensite), and γ-Fe (residual austenite). The only result of the temperature change was the difference in the proportions of each phase. The dissolution of carbides during quenching leads to increases in all the elements in the matrix. On the one hand, some alloying elements with a larger radius than Fe (such as W and Mo) cause lattice deformation and increase the lattice constant of α-Fe. On the other hand, these alloying elements can strengthen and improve the stability of austenite, which leads to a decrease in the MS temperature [29]. Therefore, with the increase in the quenching temperature, the α-Fe peak gradually shifted to the left, and the peak intensity gradually decreased. In contrast, the γ-Fe peak intensity increased, as shown in Figure 7b.

As presented in Figure 8, almost all alloying elements were uniformly distributed in the α-Fe. The chemical formula of white primary M_6_C carbides varied from Fe_3_W_3_C (or Fe_3_Mo_3_C) to Fe_4_W_2_C (or Fe_4_Mo_2_C) depending on the chemical composition [30], which mainly comprised Fe, W, and Mo. The gray primary MC carbides were mainly composed of V, Mo, and W. However, in contrast to the uniform distribution of the alloying elements in the primary M_6_C carbides, there were, respectively, rich regions of V elements and of W and Mo elements in the primary MC carbides. This difference may have been due to the formation process and physical properties of the primary MC carbides, such as the PM-HSS microstructure, which often arises under rapid solidification conditions [31,32].

The evolution of the morphologies of the carbides at different quenching temperatures and their statistical results are shown in Figure 9 and Table 1. Figure 9a–f reveal that, with the increase in temperature, the chain-like primary carbides were gradually separated and became spherically distributed in the matrix. A decreasing trend in the number of carbides was also observed. However, the carbides became aggregated and coarsened when the temperature reached 1250 °C. During the quenching process, the evolution of carbides can be divided into the following two stages.

(1) The dissolution of carbides. When an alloy is heated to the austenitizing temperature, the solubility and diffusion coefficient of the alloying elements in the austenite are increased, leading to the dissolution of carbides. At this stage, the thermodynamic and kinetic driving forces of carbide dissolution result from the reduction in the interfacial energy of the matrix/carbide and the diffusion of alloying elements from carbide into austenite, respectively. Consequently, the dissolution preferentially occurs at a large-curvature carbide/austenite interface, such as the sharp corners and depressions of the carbide [33]. When the equilibrium concentration of austenite is reached, the carbide evolution moves into the second stage.

(2) The spheroidization and re-precipitation of carbides. The following formula can be used to calculate the equilibrium concentration of alloying elements at the carbide/austenite interface [34]:(1)$$C_{\rho 1}-C_{\rho 2}=2{\Omega C}_{0}\sigma \;(1/r_{1}-1/r_{2})\;/\mathrm{RT},$$
where $C_{\rho 1}$, $C_{\rho 2}$, and $C_{0}$ are the equilibrium concentrations of alloying elements at the carbide/austenite interface with curvatures of ρ1, ρ2, and 0, respectively; σ is the austenite/carbide interface; Ω is the partial molar volume; and R is the universal gas constant. Moreover, T is the absolute temperature, and $r_{1}$ and $r_{2}$ are the radii of the curvature of carbide at the carbide/austenite interface with curvatures of ρ1 and ρ2, respectively. According to Equation (1), when austenite reaches saturation, the concentration of alloying elements at the large-curvature interface is much higher than that at the small-curvature interface. The presence of the concentration difference promotes the diffusion of alloying elements from the large-curvature interface to the small-curvature interface. Consequently, at a large-curvature interface, the austenite becomes unsaturated, and carbides continue to dissolve. However, the austenite supersaturated at the interface with a small curvature reaches the equilibrium concentration state at the respective temperature in the form of precipitated carbide. With the passage of time, the carbides gradually separate and spheroidize.

The data analysis of carbides presented in Figure 9g–i and Table 1 reveals the following. The particle size distribution of the carbides exhibited no obvious change in the quenching temperature range of 1130–1230 °C. It is worth noting that the dissolution behaviors of the primary M_6_C and MC carbides presented in Figure 9h were significantly different. With the increase in the quenching temperature, the percentage proportion of the primary M6C carbide exhibited a relatively obvious downward trend. However, for the primary MC carbide, the downward trend in the percentage proportion occurred when the temperature reached or exceeded 1230 °C.

The diffusion coefficient of the alloying element determines the dissolution of carbides [35]. As the diffusion coefficient of carbon is much higher than that of alloying elements, the dissolution of primary M_6_C and MC carbides is mainly controlled by the diffusion rates of the W, Mo, and V alloying elements. Equation (2) can be used to calculate the diffusion coefficients of these alloying elements in austenite [18]:(2)$$D=D_{0}\exp\left(-Q/\mathrm{RT}\right),$$
where D is the diffusion coefficient, D_0_ is the diffusion constant, Q is the diffusion activation energy, R is the gas constant, and T is the temperature. In austenite, the diffusion activation energies of W, Mo, and V are Q_W_ = 261.5 kJ/mol, Q_Mo_ = 247 kJ/mol, and Q_V_ = 293 kJ/mol [36], respectively. The diffusion coefficients of W, Mo, and V in austenite as a function of temperature are shown in Figure 9i. With the increase in the temperature, the diffusion coefficients of W and Mo exhibited an exponentially increasing trend, while that of the V element only increased slightly when the temperature exceeded 1200 °C. Accordingly, the area percentage of primary M_6_C carbide shown in Figure 9h exhibited an obvious decreasing trend as the temperature increased. The percentage of primary MC carbide only decreased slightly when the temperature increased to 1230 °C.

#### 3.2.2. Hardness Change and Carbide Precipitation Behavior during Tempering

The quenching temperature for tempering according to the relatively fine carbide particle distribution was selected, and the hardness curve is presented in Figure 10. For comparison, experimental data published by Mesquita [23] are introduced. With the increase in the quenching temperature, the tempering hardness of the three kinds of experimental steels exhibited an increasing trend, and the experimental data collected in this study were similar to the previously reported results. However, it is worth noting that the hardness of spray-formed HSS was obviously lower than that of CF-formed HSS when the quenching temperature was below 1220 °C.

The samples quenched at 1230 °C and tempered at 560 °C were selected for TEM observation, and the experimental results are shown in Figure 11. As can be seen from Figure 11a,b, there were many precipitates a needle-like distributions. They were about 15–20 nm in length and 2–3 nm in thickness, and the axes of the needles were aligned along identical directions, suggesting that the precipitations were generally precipitated according to a certain orientation relationship with the ferrite. A high-resolution TEM (HRTEM) image of these precipitates is shown in Figure 11c. In addition, as exhibited in Figure 11d, fast Fourier-transform (FFT) analysis was carried out for the long, needle-like precipitates shown in the boxed area of Figure 11c. Due to the small size of the precipitates, the brighter diffraction spots obtained with the FFT indicated the crystal plane of [011]α. Diffracted streaks appeared in the diffracted spots along the <100>α direction, which was caused by the difference between the atomic arrangements of the two sides of the coherent interface, resulting in elastic distortion near the phase boundary when the long needle precipitated along the (010)_α_ plane of the matrix [37,38]. The corresponding diffraction patterns shown in Figure 11d,e indicated that the precipitates were secondary M_2_C carbides with an HCP crystalline structure and were related to the ferrite lattice through the Pitsch–Schrader orientation; i.e., (-1102)_M2C_//(020)_α_, [1-101]M_2_C//[001]_α_. The (11-02)_M2C_ diffraction spots were parallel to (020)_α_ and streaked in a direction perpendicular to {020}_α_. This indicated that the long rod axis or growth direction was <-1102>_M2C_//<100>_α_. In Figure 11f, it can be seen that the -1102)_M2C_ spacing was measured as 0.1744 nm, which was close to the standard value (0.1753 nm) of the PDF card (#35-0787). In addition to secondary needle-like M_2_C carbides, there were also some granular precipitates with a diameter of 5–8 nm, as shown in Figure 12a,b. The diffraction pattern shows that these were secondary MC carbides with an fcc crystalline structure that were related to the ferrite lattice through the B-N orientation; i.e., (1-10)_MC_//(−100)_α_, [010]α//[110]_MC_ [39]. The existence of these two precipitates was the main reason for the secondary hardening effect and the hardness improvement in the HSS after tempering [40,41].

## 4. Discussion

### 4.1. Solidification Process and Formation of Segregation in SF

SF mainly consists of atomization and deposition stages (Figure 13a). During the atomization stage, atomized gas with a high cooling rate breaks the metal liquid stream into metal droplets with uniform compositions and different size distributions (10–500 μm). Due to the difference in the cooling rate caused by the size effect, the droplets can be divided into liquid droplets, partially liquid droplets, and completely solid droplets according to their size before reaching the deposited billet. The research results published by Lee [26] showed that large droplets are composed of a coarse dendrite and eutectic carbide network, while small droplets have a continuous and regular submicron carbide cell structure and exhibit the characteristics of a rapid solidification structure (Figure 13b). During deposition, when droplets arrive at the substrate, they undergo collision and fusion to form the deposited layer. As the process continues, the height of the deposited billet increases gradually. The surface of the billet forms a mushy zone and enters the steady-state stage. The mushy zone must have an equilibrium temperature above the solidus temperature, which is constant during the deposition process [21]. Under such circumstances, the solid–liquid equilibrium of the temperature in the mushy zone is maintained by the different reactions of the droplets deposited on the surface of the mushy zone. The completely liquid and partially liquid droplets cool down to Ts and solidify. Meanwhile, the completely solidified droplets are heated to Tmushy, remelt, and lose their rapid solidification characteristics (Tliquid > Tsolid/liquid ≈ Tmushy > Tsolid). With the constant input of heat as the droplet is deposited, the solidification of the remaining liquid takes place under near-equilibrium conditions when the deposition zone is cooled (Figure 13c) [14,22]. However, this model does not consider the transient deposition period, and the microstructure evolution is here described from the point at which a constant surface temperature was reached.

As shown in Figure 5 and Figure 6, the morphological change in the carbides from strip to block indicated that the cooling rate along the diameter direction was obviously affected by the heat transfer between the deposited billet surface and the atomized gas. Therefore, it can be speculated that there is a certain moment at the deposition stage when the atomized droplets do not act on the mushy area (there is no heat input or impact on dendrites). At this time, due to the external deposition surface contacting the atomized gas, solidification advances from the surface to the center and grows in the columnar dendrites under the influences of the directional heat transfer in the radial direction and the temperature gradient. Under the effect of selective crystallization, the solute elements are continuously enriched at the solidification front, and the ends of the columnar dendrites gain a higher solute concentration [42]. Moreover, the partially solidified or incompletely remelted dendrites from droplets form equiaxed morphologies due to radial heat loss at the center of the mushy zone [43]. The formation of equiaxed grains hinders the growth of columnar dendrites. Therefore, the enriched solute elements are retained at the interface between the columnar and equiaxed crystals. Segregation with severe element enrichment is formed at 1/2*R* in larger ingots in which the temperature gradients at the center and the radius are similar (Figure 13d). After that, when the droplet reaches the surface of the mushy zone, it acts on the solidified dendrite and undergoes crushing, remelting, and other reactions, causing the temperature of the mushy zone to rise to the steady-state temperature again. This situation always occurs on the surface of the mushy zone with the growth of a deposit billet. In the region below the mushy region, the liquid phase still exists locally because of slow cooling. The resistance of the interdendritic segregation liquid flowing into the liquid phase region is lower than the flow in other directions [44]; therefore, under the influence of centrifugal force (substrate rotation), gravity, density, and other factors, the interdendritic liquid overcomes the friction force imposed by the dendrites and flows into the liquid region, leading to the increase in the carbon and alloying element contents in the liquid region. The rotation of the substrate intensifies the temperature fluctuation and remelting at the solid–liquid interface, which may cause the segregation morphology to present as an annual ring (Figure 13e).

### 4.2. Discussion of the Relatively Low Hardness

The essence of secondary hardening in the tempering stage is the precipitation hardening effect caused by the secondary carbides. In general, the volume fraction, size, and type of secondary carbides determine the maximum hardness after tempering. For the M3 HSS studied in this research, according to the characterization results presented in Figure 11 and Figure 12, the precipitation strengthening of the secondary carbides in the tempering stage was mainly that of secondary MC and M_2_C carbides. The literature shows that the main components of these two precipitates are Cr, Mo, and V [45].

Figure 9 and Figure 10 present the evolution of carbide dissolution and hardness with the increase in the quenching temperature. With the increase in the quenching temperature to 1230 °C, the hardness significantly increased after tempering when the percentage of primary MC carbide decreased. This may have been due to the dissolution of primary MC carbides at high temperatures, which provides the matrix with sufficient amounts of the elements required for carbide formation during tempering [46].

As mentioned previously, solid particles formed by atomization have the characteristics of rapid solidification. Under the condition of rapid cooling, primary MC carbides are formed through the consumption of primary M_2_C carbides. In Lee’s research [26], the solid phase of the particles was found to be mainly composed of primary MC carbides and α/γ-Fe. Moreover, the proportions of the Mo, W, and Cr alloy elements in primary MC carbides increase due to the increase in the cooling rate, and the phenomenon of a non-uniform composition distribution occurs [47,48]. Although the solid particles remelt when they reach the mushy zone, significant amounts of primary MC carbides can remain in the mushy zone due to their relatively stable characteristics. Therefore, some research has found that the SF process is characterized by a higher proportion of primary MC carbides than the traditional process, and primary MC carbides have higher proportions of W, Mo, and Cr under rapid cooling conditions [49]. When the conventional heat treatment (1180 °C quenching + 560 °C tempering) is used, the primary MC carbides cannot be fully dissolved, which decreases the C, Cr, Mo, and V contents in the solution and, thus, the secondary hardening capability, leading to the hardness being lower than that of CF-formed HSS.

Based on the preceding discussion, it can be understood that the problems of segregation and hardness are caused by the particularity of SF’s combination of atomization and deposition processes. The characteristics of SF can be summarized as follows. First, it is clear from the following two points that the SF process cannot be simply considered a rapid solidification technology: (1) the solid particles formed in the atomization stage remelt in the mushy region and lose the characteristics of rapid solidification; (2) the slow cooling rate and liquid flow during the deposition stage tend to cause segregation. Second, the effect of the high cooling rate promoted during the atomization stage may be preserved in the final microstructure. Although the hardness of spray-formed M3 HSS obtained in this study was low due to the formation of a higher proportion of primary MC carbides in the atomization stage with a high cooling rate, the relevant references show that the atomization stage has great application value in relation to the wear resistance of materials [13,14,19,50]. The effects of atomization and deposition on microstructure transformation should be considered when materials are prepared using SF. This may significantly facilitate the improvement of the material properties.

## 5. Conclusions

(1)Two segregation morphologies were found in the macrostructure of spray-formed M3 HSS: (1) annual ring segregation, which was mainly located in the region from the edge to 1/2*R* and reflected the enrichment of C, W, Mo, and Cr; and (2) ring segregation, which was located at 1/2*R* and reflected the enrichment of C, W, Mo, Cr, and V. From the edge to the center of the samples, the morphologies of carbides changed from chain to block, and the carbides in the segregation region were coarser;(2)During the deposition stage, the slow cooling rate and the heat dissipation mode of the billet surface in contact with the atomized gas were found to be the fundamental reasons for the formation of segregation and the change in the carbide morphology;(3)The low hardness of spray-formed M3 HSS was due to the higher proportion of primary MC carbides that resulted from the faster cooling rate in the atomization process. As a result, the primary MC carbides could not be fully dissolved, which decreased the C, Cr, Mo, and V contents in the solution and, thus, reduced the secondary hardening capability;(4)In the context of preparing materials with large diameters, the microstructure characteristics of M3 spray-formed HSS can be determined through both atomization and deposition processes.

## Figures and Tables

**Figure 1 materials-16-00482-f001:**
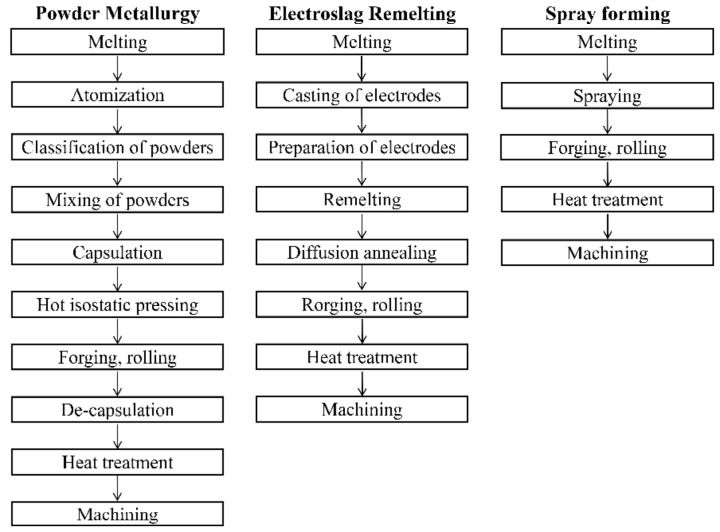
The process steps of powder metallurgy, electroslag remelting, and spray forming [14].

**Figure 2 materials-16-00482-f002:**
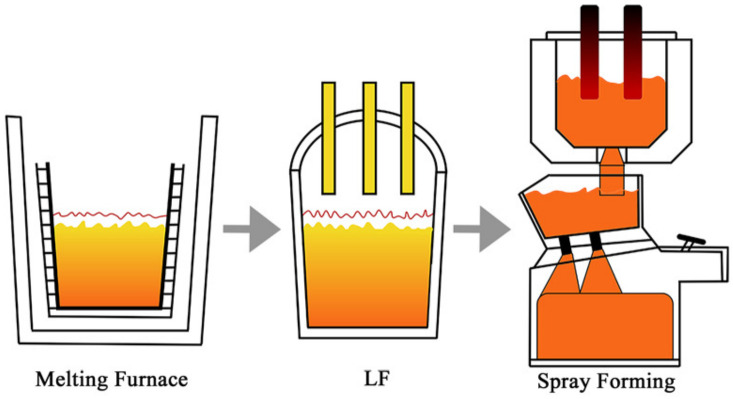
The production process for spray forming.

**Figure 3 materials-16-00482-f003:**
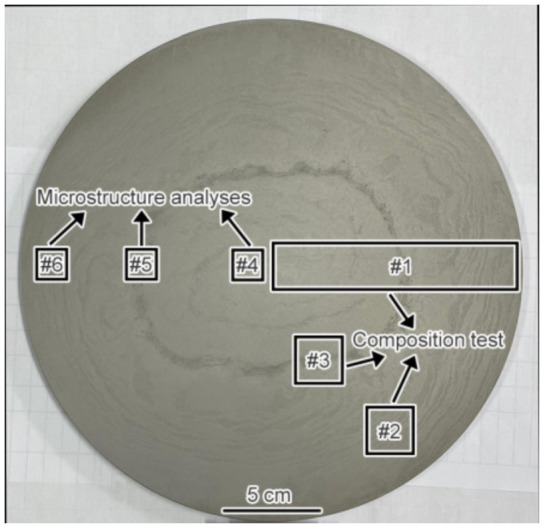
The macrostructure of the experimental steel and the schematic diagram of sampling.

**Figure 4 materials-16-00482-f004:**
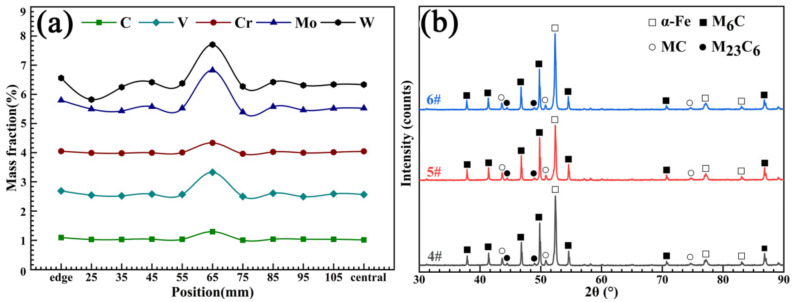
(**a**) The composition test of sample #1 and (**b**) the XRD patterns of samples #4, #5, and #6.

**Figure 5 materials-16-00482-f005:**
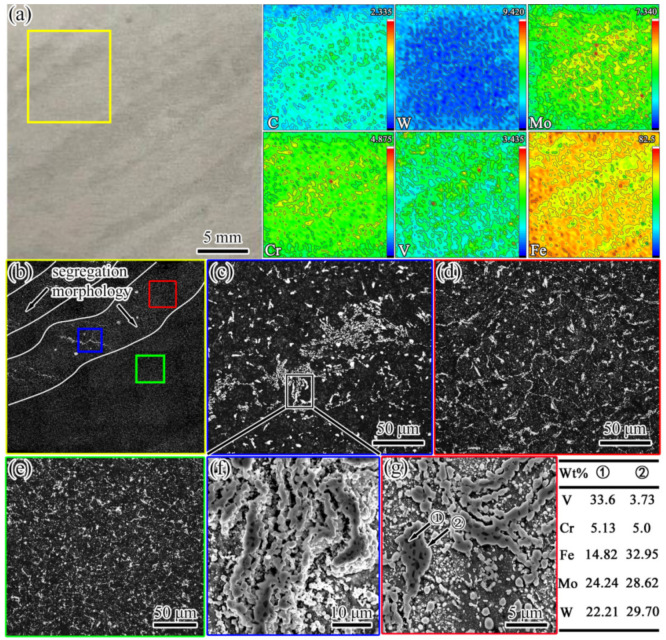
Images from the composition test and microstructure observations of sample #6. (**a**) The segregation morphology and alloying element distribution; the values in the top-right corner of each figure indicate the maximum concentration (red); (**b**) the microstructure (OM) of the area in the yellow box in (**a**); (**c**–**e**) the microstructures (OM) in the blue, red, and green boxes in (**b**), respectively; (**f**) the SEM image of carbide segregation bands corresponding to (**c**); (**g**) the SEM image of the network-like carbide.

**Figure 6 materials-16-00482-f006:**
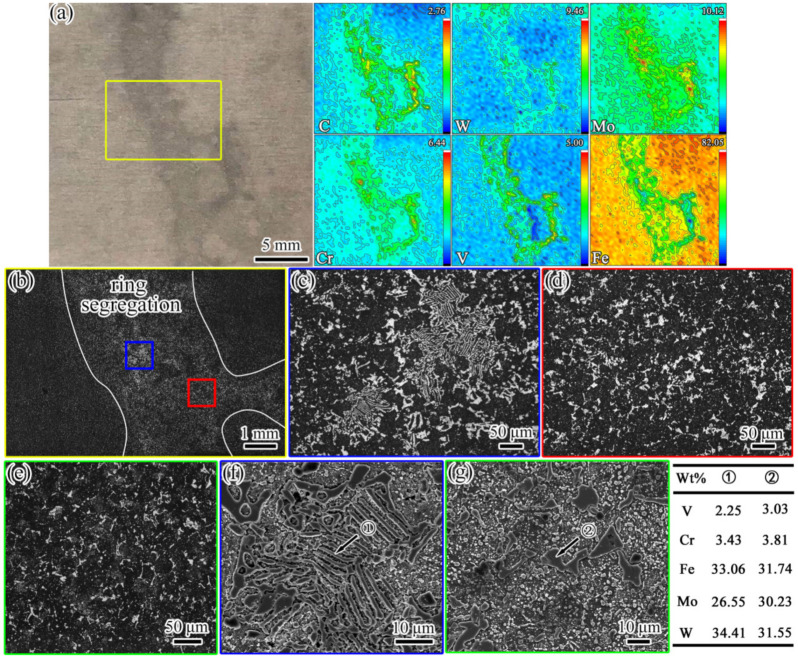
Images of composition and microstructure. (**a**–**d**,**f**) Sample #5; (**e**,**g**) sample #6. (**a**) The segregation morphology and alloying element distribution; the values in the top-right corner of each figure indicate the maximum concentration (red); (**b**) the microstructure (OM) of the area in the yellow box in (**a**); (**c**,**d**) the microstructure (OM) in the blue and red boxes in (**b**), respectively; (**e**) the microstructure (OM) of sample #4; (**f**) the SEM image of the fish bone-like carbide corresponding to (**c**); (**g**) the SEM morphology of the massive carbide.

**Figure 7 materials-16-00482-f007:**
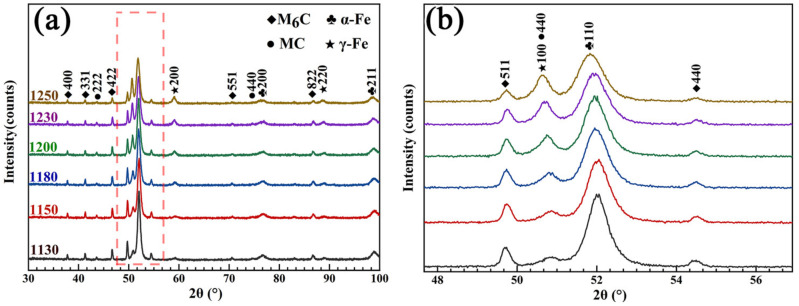
XRD patterns. (**a**) The different quenching temperatures; (**b**) partially enlarged graph.

**Figure 8 materials-16-00482-f008:**
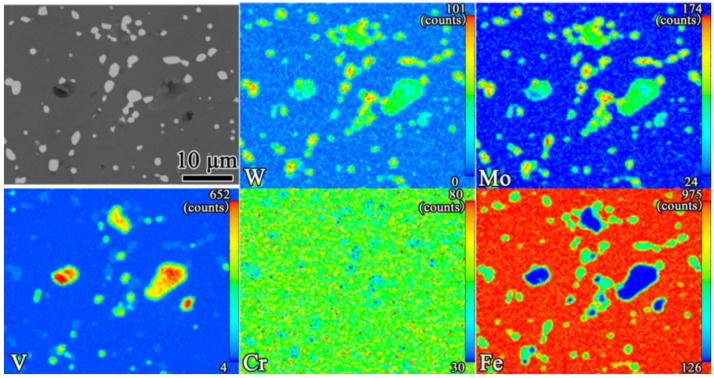
The EPMA mapping results for the quenching of samples at 1200 °C. The values in the top-right corner of each figure indicate the maximum concentration (red).

**Figure 9 materials-16-00482-f009:**
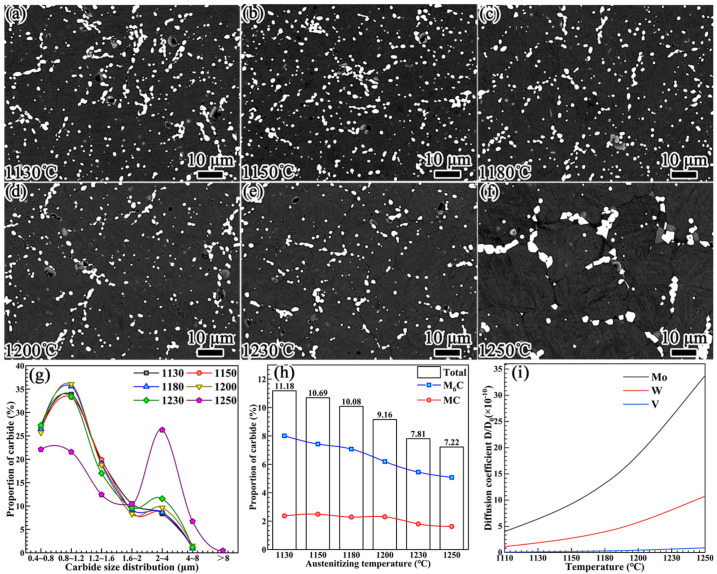
The evolution of carbides at different quenching temperatures: (**a**) 1130 °C, (**b**) 1180 °C, (**c**) 1230 °C, (**d**) 1250 °C, (**e**) 1230 °C, and (**f**) 1250 °C. (**g**) The percentages for the carbide particle size distribution. (**h**) Percentages for the carbide proportions. (**i**) Diffusion coefficients of Mo, W, and V elements as functions of temperature.

**Figure 10 materials-16-00482-f010:**
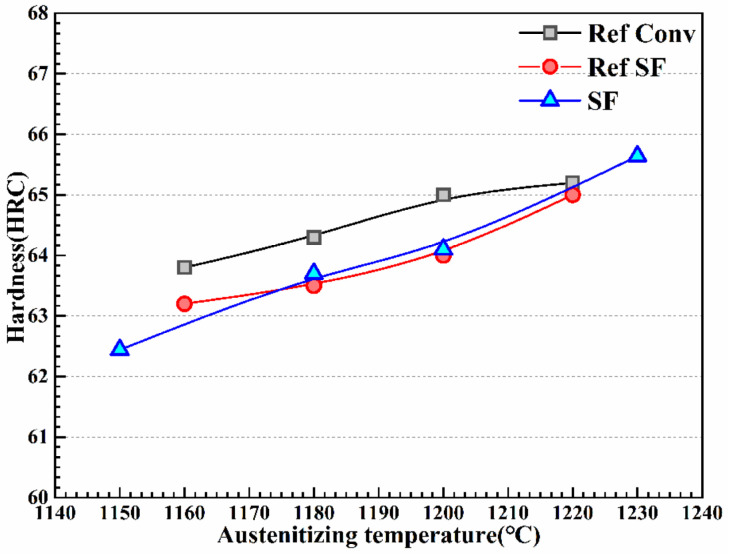
The tempering hardness curve at different quenching temperatures.

**Figure 11 materials-16-00482-f011:**
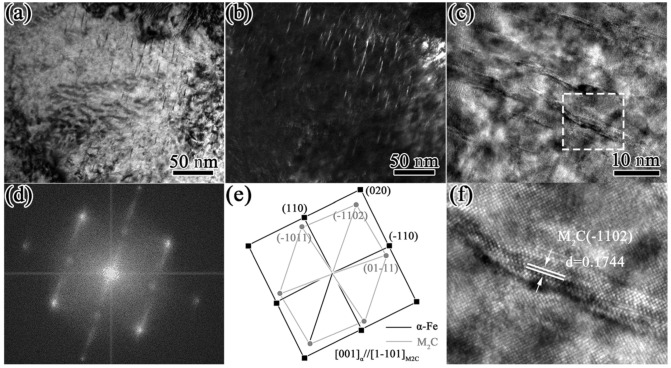
TEM images of short, rod-like precipitates: (**a**) bright field image; (**b**) dark field image; (**c**) high-resolution TEM image. (**d**,**e**) The Fourier pattern; (**f**) IFFT image of rectangular area in (**c**).

**Figure 12 materials-16-00482-f012:**
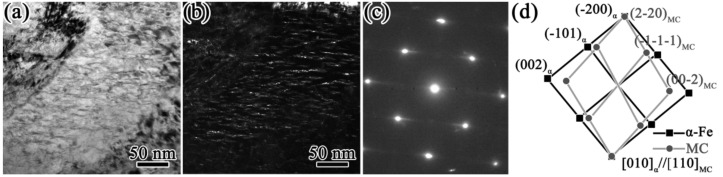
TEM images of granular-like precipitates. (**a**) bright field image; (**b**) dark field image; (**c**,**d**) diffraction pattern.

**Figure 13 materials-16-00482-f013:**
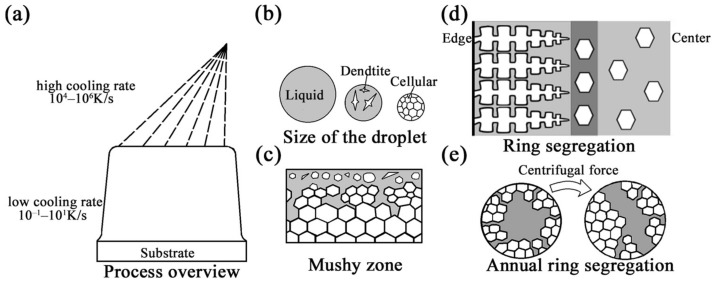
The solidification process and formation of segregation during the SF process. (**a**) Overview of the spray-forming process; (**b**) the states of atomized droplets with different sizes; (**c**) the mushy zone; (**d**) the formation of ring segregation; (**e**) the formation of annual ring segregation.

**Table 1 materials-16-00482-t001:** The carbide quantity statistics for different quenching temperatures.

Process/Temperature	1130 °C	1150 °C	1180 °C	1200 °C	1230 °C	1250 °C
Amount (*n*)	5108	4999	4939	3955	3432	1650

## Data Availability

Not applicable.

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
