# Peer review of "Discussion of the Segregation and Low Hardness of Large-Diameter M3 High-Speed Steel Produced by Spray Forming"

_materials, 2023, doi:10.3390/ma16020482_

Round 1
Reviewer 1 Report
This paper is well written from language and technical point of view. However, few grammatical errors exist for example line no 16 of abstract need to be revised and to be checked thoroughly. Also, abstract presents a relevant problem statement but seems to be very detailed resulting in confusion please make it little brief with generic concluding remarks.
The design of the study and methodology is consistent with the objectives of the study and adding scientific value to the existing knowledge base. All the pictorial representations, statistical results and graphs are clear and justifying the findings (except the XRDs need to be sharper and clearer visibly). However, the results should be discussed incorporating latest reference papers of the last 5 years to gather the latest information on the topic area of the literature. Conclusion is effectively reflecting the findings and overall, this paper would be a scientifically value adding piece of research.
Author Response
Question 1: abstract presents a relevant problem statement but seems to be very detailed resulting in confusion please make it little brief with generic concluding remarks.
The abstract has been modified:
Abstract: As an advanced near-net-shape processing method in which directly preformed semi-finished products are created from liquid metals, spray forming has become popular in the development and application of new materials, and is supporting industrialization. However, as investigated in this work, the problems of segregation and low hardness exist in the actual industrialization process, particularly for large-diameter M3 high-speed steel. It is found that the annual ring segregation morphologies are mostly distributed from the edge to 1/2R, with a large number of stripes primarily enriched in C, Mo, and Cr elements, and the degree of segregation is mild. The ring segregation is located at the 1/2R position, where the main elemental enrichments are C, W, Mo, Cr, and V, and the segregation degree is severe. The formation of segregation during deposition is described based on an equilibrium solidification model. A slow cooling rate and heat dissipation from the surface to the inside are considered the main factors causing segregation and changes in the carbide morphology. In terms of hardness, with the increase of the quenching temperature to 1230 °C, the tempering hardness increases significantly. The analysis shows that a faster cooling rate in the atomization stage causes the solidified droplets to have rapid solidification characteristics, and there will be a higher proportion of MC carbide in the deposited billet. MC carbides cannot be fully dissolved under the conventional heat treatment process, which decreases the C, Cr, Mo, and V contents in the solution and thus reduces the secondary hardening capability. The findings show that when using the spray forming process to prepare large-diameter materials, it should not be considered a rapid solidification technology simply because of its atomization stage. Moreover, more attention should be paid to the influence of microstructure transformation during atomization and deposition.
Question 2:
XRD image pixels are all 600dpi
Question 3: the results should be discussed incorporating latest reference papers of the last 5 years to gather the latest information on the topic area of the literature.
In fact, there are few reports on the segregation and low hardness of spray formed high speed steel, which is also a relatively new research direction of this paper. Therefore, it is difficult to find relevant documents in the past five years and introduce them into the results and discussions

Reviewer 2 Report
The manuscript is on the segregation and hardness of large diameter M3 high speed steel produced by spray forming. The manuscript is well formulated, and results are interesting. Please see below for my comments.
1. “Because of its excellent wear resistance and red hardness, high-speed steel (HSS) is 45 widely used in large machining tools, machinery manufacturing, and other fields.” Provide refences for this.
2. In page 3, “After spraying 112 and deposition, the bill was annealed and finally left the factory in the form of rolled and 113 softened annealed bars.” You mean “billet”?
3. In page 4, “After quenching, the samples were 139 immediately tempered at 560 °C for 3 × 1 h and air-cooled to room temperature.” What is 3x1 h? You mean 3 hr?
4. In Fig. 4a, how this composition was measured? Please mention in text.
5. In Fig 13, describe figure (a-e) in caption.
Author Response
- “Because of its excellent wear resistance and red hardness, high-speed steel (HSS) is 45 widely used in large machining tools, machinery manufacturing, and other fields.” Provide refences for this.
References(1,2) have been added to the article
- In page 3,“After spraying 112 and deposition, the bill was annealed and finally left the factory in the form of rolled and 113 softened annealed bars.” You mean “billet”?
Modified according to expert opinions
- In page 4, “After quenching, the samples were139 immediately tempered at 560 °C for 3 × 1 h and air-cooled to room temperature.” What is 3x1 h? You mean 3 hr?
Modified as: After quenching, the sample shall be tempered at 560 ° C for 1 hour, and air cooled to room temperature, and repeated three times
- In Fig. 4a, how this composition was measured? Please mention in text.
In section 2.2 of the article, the experimental scheme and method have been mentioned
- In Fig 13, describe figure (a-e) in caption
Modified as required by experts
Reviewer 3 Report
Publishing a paper on the theoretical basis of segregation needs to get deeper into the thermodynamics of diffusion, reactions, and solidification. On the other hand, in this article, a solution to reduce segregation is not clearly presented.
Author Response
We are sorry that we have not yet solved the problem of segregation of large-sized spray formed high speed steel. In this paper, the particularity of spray forming process and some views on the process itself are discussed.

Round 2
Reviewer 1 Report
The author have addressed the comments.
It is ok to publish now.
Author Response
Thank the experts for their suggestions
Reviewer 3 Report
Although the authors have reported many high-quality results, there is no basic idea and final output for eliminating segregation.
The paper must be rewritten by a material expert. For example, in the following sentence, they should use "etching" instead of "corrosion...":
"A nitric acid solution with a volume ratio of 1:10:10 was used for the corrosion of the macrostructure"
As I mentioned earlier, publishing a paper on the theoretical basis of segregation needs to get deeper into the thermodynamics of transformations (diffusion, reactions, and solidification).
However, this can be considered a kind of report paper, and the discussion can be used by other researchers in the future.
Author Response
The article has been revised
